# Study on Single-Tree Segmentation of Chinese Fir Plantations Using Coupled Local Maximum and Height-Weighted Improved K-Means Algorithm

**Xiangyu Chen** [1], **Kunyong Yu** [1,*], **Shuhan Yu** [1,2], **Zhongyang Hu** [1], **Hongru Tan** [1], **Yichen Chen** [1], **Xiang Huang** [1] **and Jian Liu** [1]

1 College of Forestry, Fujian Agriculture and Forestry University, Fuzhou 350002, China; 1210454001@fafu.edu.cn (X.C.); 2220430007@fafu.edu.cn (S.Y.); 2220430006@fafu.edu.cn (Z.H.); 3210422062@fafu.edu.cn (H.T.); 3210422008@fafu.edu.cn (Y.C.); 1210430004@fafu.edu.cn (X.H.); fjliujian@fafu.edu.cn (J.L.)
2 Fujian Chuanzheng Communications College, Fuzhou 350002, China
* Correspondence: yuyky@fafu.edu.cn

**Abstract:** Chinese fir (*Cunninghamia lanceolata*) is a major timber species in China, and obtaining and monitoring the parameters of Chinese fir plantations is of great practical significance. With the help of the K-means algorithm and UAV-LiDAR data, the efficiency of forestry surveys can be greatly improved. Considering that the traditional K-means algorithm is susceptible to the influence of initial cluster centers and outliers during the process of individual tree segmentation, it may result in incorrect segmentation. Therefore, this study proposes an improved K-means algorithm that uses the methods of local maxima and height weighting to optimize and improve the algorithm. The research results are as follows: (1) Compared to the traditional K-means algorithm, the producer accuracy and user accuracy of this research algorithm have imsproved by 10.72% and 11.46%, respectively, with significant differences ($p < 0.05$). (2) The research algorithm proposed in this study can adapt to Chinese fir plantations of different age groups, with average producer accuracy and user accuracy reaching 78.48% and 83.72%, respectively. In summary, this algorithm can be effectively applied to the forest parameter estimation of Chinese fir plantations and is of great significance for sustainable forest management.

**Keywords:** single tree segmentation; LiDAR; K-means; point cloud; local maximum





## 1. Introduction

Forest resources are important supports for terrestrial ecosystems and play a crucial role in ecological environment governance and protection [1,2]. Periodic, large-scale monitoring and investigation of forest resources can contribute to their protection and management. Compared to traditional forestry surveys, unmanned aerial vehicle (UAV) remote sensing technology can be used to monitor forest resources over short periods, over large areas, and with a high level of accuracy. The recent development of unmanned aerial vehicle light detection and ranging (UAV-LiDAR) has brought about new changes in forest resource surveys. Although LiDAR does not provide rich spectral information, compared with traditional optical remote sensing, it can obtain three-dimensional structural information on the detected target. This is conducted by relying on the Differential Global Positioning System (DGPS) and the Inertial Navigation System (INS) carried by the LiDAR. This means that it can provide single-tree structural parameters, such as tree height and crown width, for forest resource surveys without damaging the forest stand. The single-tree segmentation method can be used to separate several pieces of individual tree information within a large-scale LiDAR dataset, which is a prerequisite for obtaining single-tree structural parameters.

Current research on airborne LiDAR-based single-tree segmentation can be divided into single-tree segmentation based on the Canopy Height Model (CHM), and single-tree segmentation based on point clouds and voxel [3–7]. Most single-tree segmentation methods based on CHM use watershed algorithms, multiscale segmentation, region seed growing algorithms, and their derivative algorithms to achieve segmentation [8–10]. This method does not operate directly on point-cloud data. Although this reduces the complexity of data processing, part of the three-dimensional structural information is lost.

The single-tree segmentation method that directly acts on point clouds and voxels typically uses the spatial aggregation of a point cloud and relies on clustering methods such as Density-Based Spatial Clustering of Applications with Noise(DBSCAN), Point cloud segmentation(PCS),K-means, Mean-shift, and Gaussian mixture models to realize the segmentation of a single-tree point cloud. Clustering is a data analysis method that automatically divides samples into several categories by measuring feature similarities or distances [11]. It belongs to unsupervised classification learning, that is, unsupervised learning. The model is not affected by data, data volume, or other information for autonomous learning. The dataset is then divided to obtain the clustering result. A classic clustering algorithm is the K-means algorithm. Following its proposal by Steinhaus [12] in 1955, K-means has been widely used in various research fields owing to its strong applicability. Among these, the K-means algorithm is a classic clustering method. Since its proposal by Steinhaus in 1955, the K-means algorithm has been widely used in various research fields because of its strong applicability. The K-means algorithm has strong applicability and high relative scalability when handling large amounts of data [13]. When the boundaries of various types of data are clear, the K-means algorithm usually obtains strong classification results. Although K-means has strong applicability and stability, it also exposes the following defects in the application process. (1) The K-means algorithm randomly selects the initial cluster center positions, which can easily cause the computation results to converge to a local optimal solution. (2) The K-means algorithm requires setting the number of clustering categories K manually, which has a high degree of subjectivity. (3) The K-means algorithm is sensitive to outliers, and outliers in the dataset affect the position of the cluster center during the clustering process. To some extent, this can lead to biased clustering results. To address these issues, this study proposed a highly weighted K-means algorithm to optimize the traditional method. In this study, an algorithm was applied to perform individual tree segmentation on different age groups of Chinese fir (*Cunninghamia lanceolata*) plantations on the Yangkou state-owned forest farm in Shunchang County. The segmentation results were compared with those obtained using the traditional K-means method to verify the impact of the improved algorithm on the results of individual tree segmentation.

## 2. Materials and Methods

### 2.1. Study Area

The study area is located in the Yangkou state-owned forest farm in Shunchang County, Fujian Province ($117°29'$–$118°14'$ E, $26°38'$–$27°121'$ N), with an average annual temperature of 18.5 °C, an average annual rainfall of 1880 mm, and a frost-free period of 305 days per year. Ten sample plots were set up for different age groups in the study area, and Chinese fir was the dominant tree species in all sample plots. The study area and sample plot locations are shown in Figure 1.

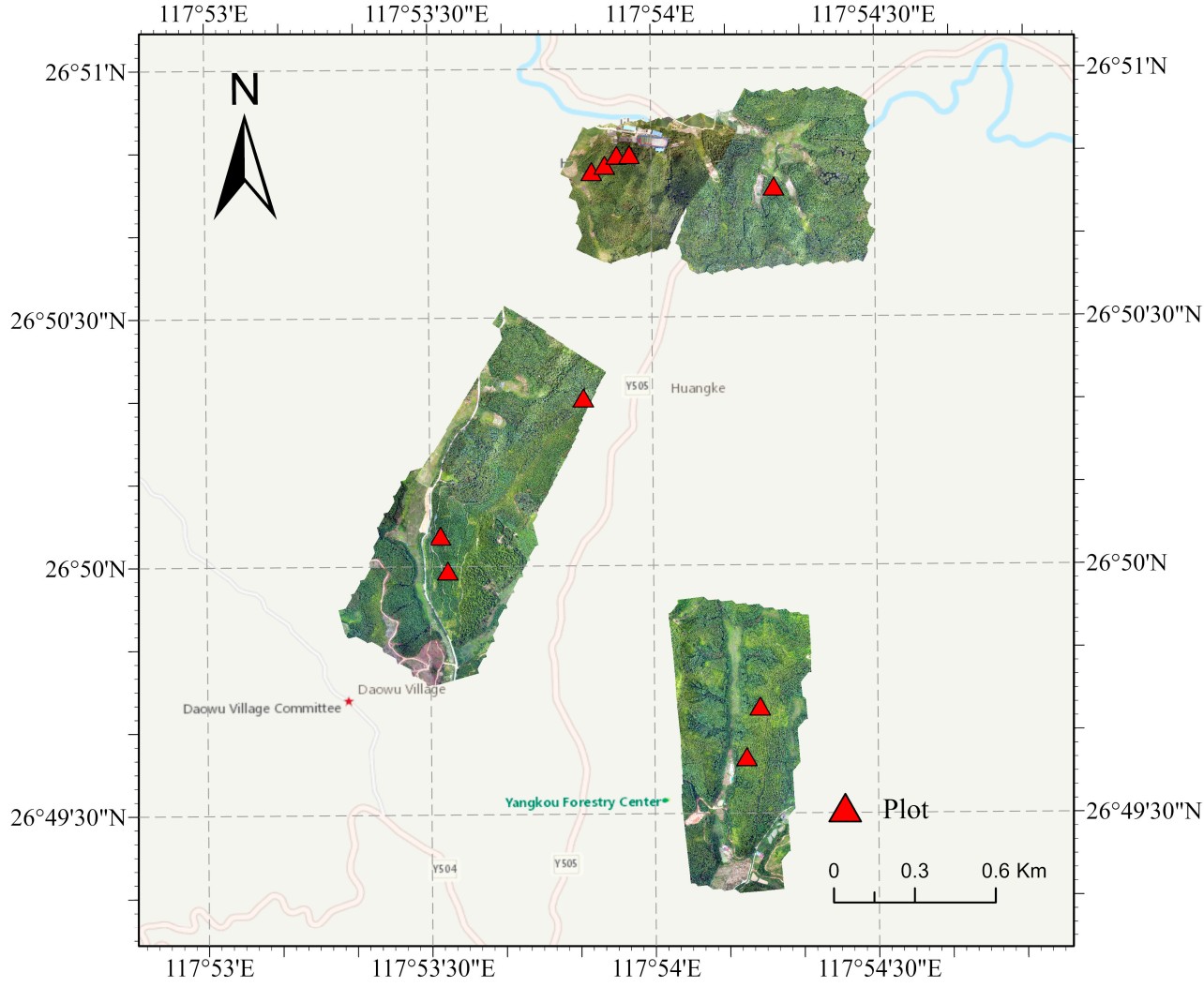

**Figure 1.** Location of the study area.

*2.2. Data*

2.2.1. Ground Survey Data Acquisition

Within the Chinese fir plantation survey area, 10 plots (25.82 m × 25.82 m) were divided according to age group. Each tree was measured to obtain information including the diameter at breast height, crown width, and tree height. The specific plot survey contents are presented in Table 1. The plots included Chinese fir plantations from five age groups, ranging from young to overmature, with canopy closure ranging from 0.6–0.9. By combining UAV LiDAR and ground survey data, visual interpretation was performed to determine the number and location of individual trees and their crown width ranges in the plots as validation data for the algorithm. The total number of reference trees measured in Plots 1–10, which served as validation data for the algorithm, is listed in Table 2.

**Table 1.** Survey information from the 10 plots surveyed during the study.

| No. | Average Tree Age | Age Group | Average DBH (cm) | Average TH (m) | Average CW (m) | Canopy Closure | Slope (°) | Aspect | Altitude (m) |
|---|---|---|---|---|---|---|---|---|---|
| 1 | 7 | Young forest | 13.0 | 8.50 | 2.60 | 0.7 | 26.00 | WN | 217 |
| 2 | 7 | Young forest | 12.6 | 9.00 | 2.50 | 0.9 | 26.00 | W | 230 |
| 3 | 12 | Middle-aged forest | 16.9 | 12.00 | 3.10 | 0.7 | 26.00 | WN | 202 |
| 4 | 12 | Middle-aged forest | 15.2 | 14.00 | 2.90 | 0.7 | 34.00 | W | 189 |
| 5 | 21 | Near-ripe forest | 18.8 | 17.00 | 3.30 | 0.9 | 28.00 | EN | 227 |
| 6 | 25 | Near-ripe forest | 19.5 | 17.00 | 3.20 | 0.6 | 30.00 | W | 255 |
| 7 | 29 | Mature forest | 19.3 | 16.00 | 3.40 | 0.7 | 28.00 | W | 210 |
| 8 | 29 | Mature forest | 18.6 | 16.30 | 3.20 | 0.8 | 34.00 | WN | 211 |
| 9 | 56 | Overripe forest | 28.9 | 22.00 | 6.80 | 0.6 | 32.00 | N | 202 |
| 10 | 56 | Overripe forest | 29.1 | 19.00 | 7.00 | 0.6 | 28.00 | EN | 196 |

**Table 2.** The total number of reference single trees in 10 plots.

| No. | Reference Total Number of Single Wood |
|---|---|
| 1 | 167 |
| 2 | 190 |
| 3 | 89 |
| 4 | 90 |
| 5 | 78 |
| 6 | 59 |
| 7 | 85 |
| 8 | 61 |
| 9 | 23 |
| 10 | 24 |

### 2.2.2. UAV-LiDAR Data

In the study area, a FEIMA Lidar UAV D500 equipped with a HESAI XT32 (Shenzhen FEIMA Robot Co., Ltd., Shenzhen, China) sensor was used to collect UAV LiDAR point-cloud data from 31 July to 8 August 2022. Owing to the numerous hills and substantial terrain fluctuations in the study area, a terrain-following flight method was used with the terrain-following height set to 150 m, scan overlap rate set at 80%, three-echo echo mode, laser level of CLASS1, and average point-cloud density of 180–230 pts/m$^2$. The LiDAR and multispectral image data are shown in Figure 2.

### *2.3. Methods*

### 2.3.1. LiDAR Point Cloud Data Preprocessing

The LiDAR data collected were corrected and stitched to obtain raw point-cloud data within the study area. Raw point cloud data collected by airborne LiDAR are affected by the surrounding environment and the characteristics of the measured targets, resulting in noise points. Denoising methods were applied to the raw point-cloud data to obtain standard point-cloud data.

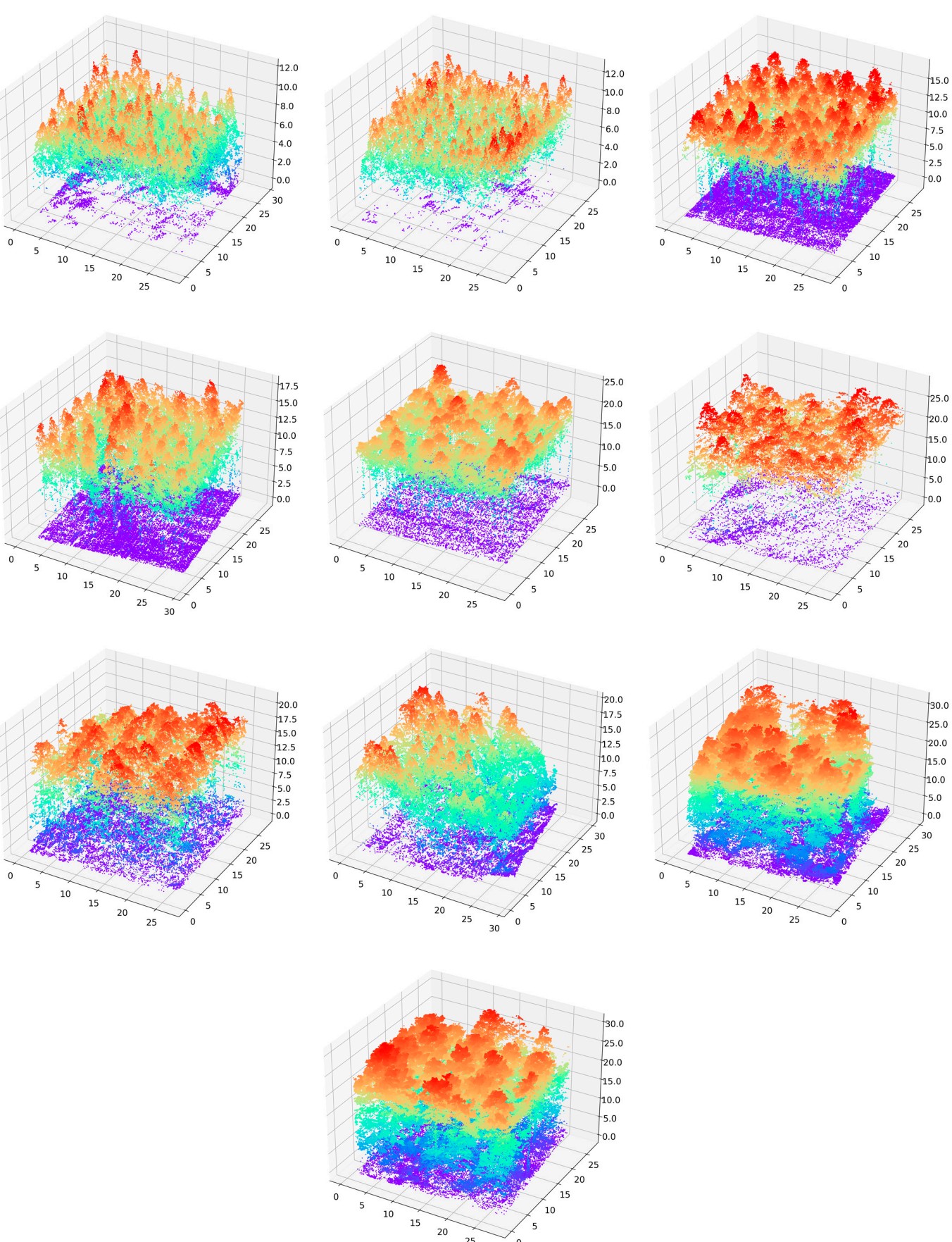

**Figure 2.** LiDAR data collected by the HESAI XT32 sensor. Note: The color from purple to red represents a rise in height.

### 2.3.2. Point Cloud Filtering

To obtain high-precision terrain data and canopy height models, it is necessary to divide airborne LiDAR point cloud data into ground and non-ground points, that is, point cloud filtering [14], which is a prerequisite for the application of LiDAR data in forest resource management. Commonly used point-cloud filtering methods include slope-based, morphology-based, and cloth filtering algorithms. Among them, the slope-based cloth filtering algorithm has a more effective performance in flat and open streets and densely built urban areas. Its performance was relatively poor in hilly areas, with substantial terrain changes and steep terrain. However, the steps of a filtering algorithm based on morphology are relatively complex, and the overall efficiency was relatively low. Given the large amount of information and data obtained by LiDAR, considerable time and storage resources are required to process them using this method. The cloth simulation filter (CSF) is a fast point-cloud filtering algorithm proposed by Zhang [15]. The core idea is to invert the collected LiDAR point-cloud data, simulate the process of a piece of cloth naturally falling from above, and cover the surface of the inverted point cloud under the influence of gravity. A fallen cloth surface is used to represent the current terrain. The cloth filtering algorithm has a high accuracy, short processing time, and requires fewer input parameters. Considering these factors, this study used a cloth filtering algorithm for point cloud filtering, dividing the ground and vegetation point clouds.

### 2.3.3. Canopy Height Model Construction

Based on the ground point cloud obtained by CSF, a digital elevation model (DEM) with a spatial resolution of 0.1 m was obtained using the Kriging interpolation method. The purpose of point cloud height normalization was to remove the influence of terrain fluctuations on the height value of the forest vegetation point cloud so that the height of the vegetation point cloud was consistent with the real vegetation height [16]. Based on the DEM and pre-processed standard point cloud, the standard point cloud was normalized by relying on the positional relationship between the standard point cloud and the DEM pixels [17], resulting in the normalized point cloud. Then, using the LAS Dataset to Raster tool in ArcGIS Pro 3.1, the normalized point cloud was rasterized to obtain the canopy height model (CHM). The parameter settings for the LAS Dataset to Raster tool were as follows: the interpolation type chose "Triangulation", the interpolation method chose "Natural Neighbor", the point thinning type chose "Window Size", the selection method chose "Maximum", the resolution chose 0.25 m, the sampling value chose 0.1 m, and other parameters were set to default values.

Owing to the high density of the original point cloud, the resolution of the CHM was relatively high. This can lead to the presence of noisy pixels with abnormal height values. This phenomenon can cause substantial differences in pixel values within the same tree crown in the CHM, thereby affecting the detection of treetop points [18]. In this study, a mean filtering method was used to smooth the CHM and eliminate the influence of abnormal points [19]. The filtering window was circular with a radius of 0.4 m.

### 2.3.4. Local Maximum Algorithm

The local maximum algorithm is commonly used for single-tree detection and location. Based on the CHM image constructed from LiDAR point cloud data, the algorithm relies on determining the maximum grayscale value within the tree crown. This corresponds to the location of the tree top point to determine the center position of the tree crown [20,21]. The local maximum algorithm used a moving window to search for the local maximum values. In this study, ArcGIS Pro 3.1 software was used to extract Chinese fir tree points. A Focal Statistics tool was used to extract the local maximum values from the filtered CHM image, with a circular attribute window shape and search radius set to half the minimum crown width of the trees in the plot. Given that low shrubs and herbs within the forest window may also be detected during the search for maximum value points, points with height values less than 0.3 m were removed to eliminate these interferences. The process of

obtaining treetop point data using the local maximum algorithm is shown in Figure 3. The red squares in the figure represent the identified treetop positions, the range formed by the green squares represents the tree crown, the yellow squares represent the search window for local maximum values, and the length of the blue arrow represents the moving step size of the window.

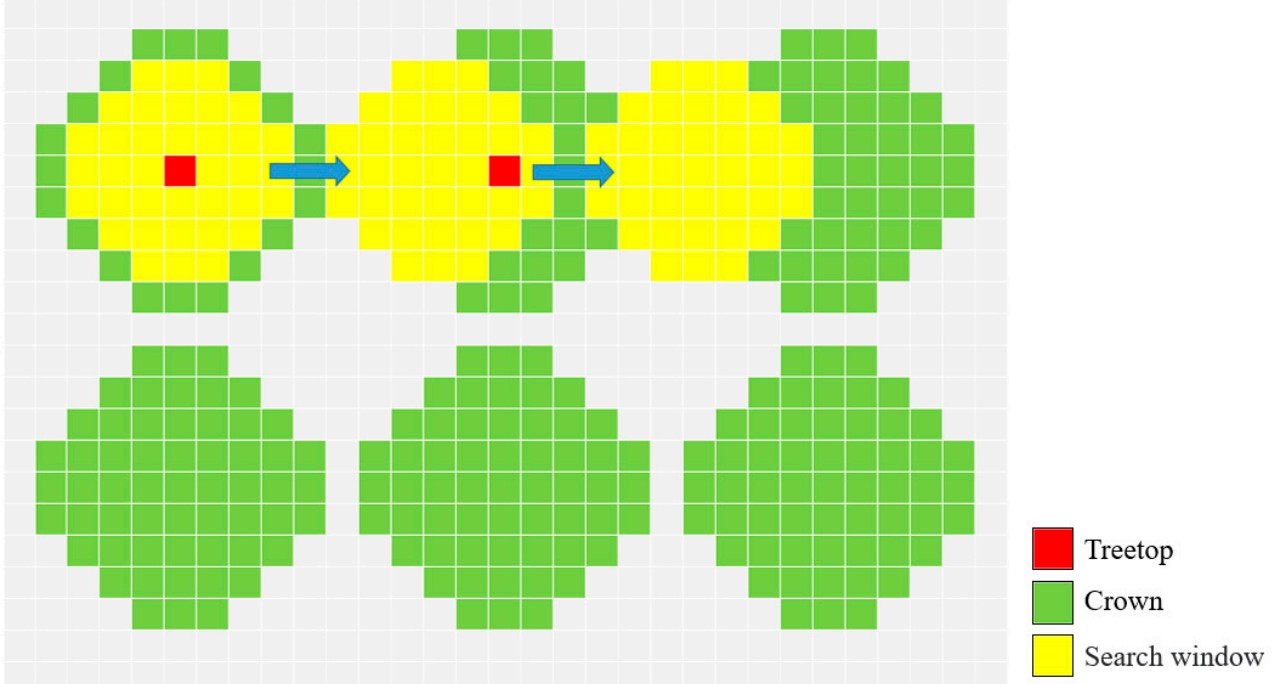

**Figure 3.** Local maximum extraction tree vertices.

### 2.3.5. Improved K-Means Clustering Algorithm

The traditional K-means algorithm requires the pre-specification of the number of partitions n, which can divide the input samples into n groups. The K-means algorithm can be divided into three steps. Firstly, select the initial cluster center by randomly selecting the sample coordinates of the dataset as the initial cluster center, and then classify each sample to the nearest cluster center. Secondly, calculate the average value of all sample points assigned to each previous cluster center to create a new cluster center. Thirdly, determine whether there was a difference between the new and old cluster centers. If there was a difference, repeat the last two steps until the cluster center no longer moves.

Although the traditional K-means method can adapt to large sample datasets, such as LiDAR point clouds, and has been used in many different fields, the K-means algorithm faces three challenges when applied to single-tree segmentation. Firstly, the position of the initial cluster center was randomly selected, which can lead to the segmentation result being trapped in a local optimal solution. Secondly, the number of cluster centers needs to be manually determined, which is subjective. Thirdly, the traditional K-means algorithm was sensitive to outliers. When calculating the cluster center using the mean value, outliers in the dataset will affect the position of the cluster center.

To address the issues of randomly selecting the initial cluster center positions and manually setting the number of clusters in the K-means method, the following approach was proposed. A retrieval window was defined with a size equal to the average canopy radius of the study area. Single-tree detection was performed within the study area to obtain the coordinate positions of the highest points of all individual trees within the window range. This provides the locations and quantities of trees in the sample plot, which correspond to the coordinate positions of the initial cluster centers and the number of clusters n in the K-means algorithm.

Secondly, to address the sensitivity of cluster center calculation to outliers, this study adopted a weighted average method that incorporates point cloud height as a clustering weight in cluster center position calculation. This method makes the cluster center calculation more sensitive to tree height, with point clouds near the treetop being assigned higher weights. As a result, the new cluster center will be closer to the treetop position, and it can also avoid the situation where the cluster center is affected by outliers and drifts towards the edge of the canopy. The equation for calculating the weighted cluster center coordinates $\overline{x}_w$ was as follows:

$$\overline{x}_w = \frac{\sum_i^n x_i h_i}{\sum_i^n h_i} \tag{1}$$

where $n$ is the total number of point clouds in the current clustering category, $i$ is the current point cloud sequence number, $x_i$ is the coordinates of the $i$-th point cloud, and $h_i$ is the height of the $i$-th point cloud.

To summarize the previous improvement methods, the improved K-means method could also be divided into three steps. Firstly, the local maximum method was used to obtain the number and coordinate positions of trees in the sample plot, which were used as the number of partitions n and the coordinates of the initial cluster centers for the point cloud dataset. Secondly, the weighted average value of all sample points assigned to each previous cluster center was calculated to create a new cluster center. Thirdly, it was still necessary to determine whether there was a difference between the new and old cluster centers, repeating steps 2 and 3 until the cluster center coordinates no longer changed.

### 2.3.6. Tree Canopy Boundary Sketching

The results of point cloud segmentation are usually presented in the form of individual tree point clouds, which leads to reliance on visual interpretation to confirm segmentation accuracy when verifying individual tree segmentation. The alpha-shape algorithm is a relatively simple and rapid method proposed by Edelsbrunner H to obtain the boundary of a point set [22]. The algorithm assumes a circle with a radius of a set on the point cloud set S, which rolls around the specified point cloud set S. When the value of radius a is sufficiently small, each point in the point-cloud set is considered a boundary point. When it has been adjusted to an appropriate threshold, the trajectory of the rolling circle is regarded as the boundary of the point cloud. Using the alpha-shaped algorithm, the 2D boundary of the single-wood point cloud can be drawn rapidly, as shown in Figure 4.

### 2.3.7. Accuracy Verification

The tree crown matching method proposed by Zhen [23] was used to verify the results of single tree segmentation. Single-tree boundary information was manually drawn based on multispectral and CHM images. The single-tree boundary obtained by the alpha-shape algorithm was regarded as the detected canopy boundary, and the manually drawn single-tree boundary was regarded as the reference canopy boundary. According to the matching rules, when the proportion of the crown area of the detected single tree and the area of the reference single tree exceeds 50%, the single-tree segmentation result matches the real result 1:1. The calculation process is shown in Figure 5.

The single-tree segmentation accuracy detection index uses user accuracy ($U_A$) and producer accuracy ($P_A$). The equations for user and producer precision are as follows:

$$U_A = \left(\frac{N_{1:1}}{N_d}\right) \times 100\% \tag{2}$$

$$P_A = \left(\frac{N_{1:1}}{N_r}\right) \times 100\% \tag{3}$$

where $N_{1:1}$ is the number of individual trees that match the segmentation and ground truth results, $N_d$ is the total number of individual trees detected, and $N_r$ is the total amount of single wood for reference.

**Figure 4.** Alpha-shaped algorithm to obtain the 2D boundary of a single tree point cloud.

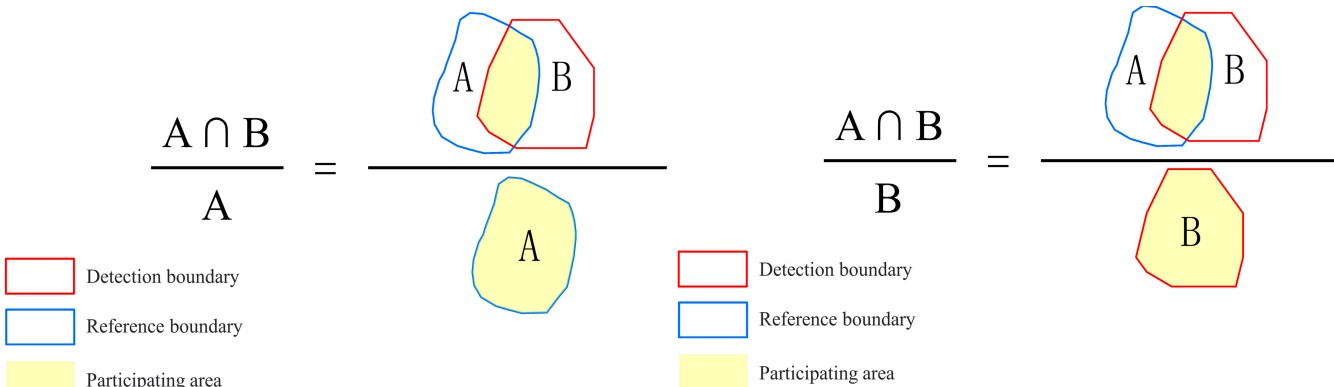

**Figure 5.** 1:1 Matching rule between the segmentation and real canopy boundaries.

### 2.3.8. Statistical Analysis

The accuracy of all single tree segmentation results was statistically analyzed using SPSS 22.0. One-way ANOVA was used to analyze the statistical significance differences in producer accuracy and user accuracy among different segmentation algorithms to examine the significant difference between the improved K-means algorithm and the traditional K-means algorithm. The coefficient of variation was calculated for the producer accuracy

and user accuracy of different single tree segmentation algorithms to analyze the stability of segmentation accuracy among different age groups.

## 3. Results

After processing the CHM of the 10 sample plots, a smoothing process was first applied to them using a window size of 0.4 m. Next, the local maximum method was used to extract the positions of the treetops in the 10 sample plots, with the search radius of the local maximum set to half of the minimum canopy width in the study area. The results of extracting treetops in the 10 sample plots are shown in Table 3.

**Table 3.** Tree points detection result information.

| No. | Detected Tree Points |
|---|---|
| 1 | 147 |
| 2 | 167 |
| 3 | 80 |
| 4 | 84 |
| 5 | 71 |
| 6 | 56 |
| 7 | 83 |
| 8 | 59 |
| 9 | 23 |
| 10 | 24 |

Based on the results from the local maximum, the K value and coordinate information of the initial cluster position were obtained, which provided the basis for the operational parameters of the improved K-means algorithm. The weighted K-means algorithm was applied to the results of the local maximum values and height-normalized point cloud data. This resulted in a single-tree segmentation of the ten sample plots. The segmentation results are shown in Figure 6. To verify the improvement of the K-means algorithm in single-tree segmentation compared to traditional single-tree segmentation algorithms, we also used the traditional K-means algorithm, watershed algorithm, and PCS algorithm to segment the individual trees in the 10 sample plots. During the calculation process, the K value of the traditional K-means algorithm was consistent with that of the improved K-means algorithm, and the seed points of the watershed algorithm were consistent with those of the improved K-means algorithm to ensure that the calculation conditions of each algorithm were consistent.

Given that the results of point-cloud segmentation were presented in the form of a set of points, it was difficult to compare and calculate the results of single-tree segmentation. Therefore, in this study, the alpha-shape algorithm was used to process the results of point cloud segmentation to obtain the concave boundary of a single-tree point cloud. The processing results are shown in Figure 7.

Based on the single-tree boundary outlined by the alpha-shape algorithm, the overlapping relationship between the detected tree crown and the real tree crown could be quantified to more effectively verify the accuracy of single-tree segmentation, calculate the user accuracy and producer accuracy of the traditional K-means algorithm, and improve the K-means algorithm, marker watershed algorithm, and PCS algorithm for single-tree segmentation results of 10 sample plots. The specific segmentation results are shown in Table 4.

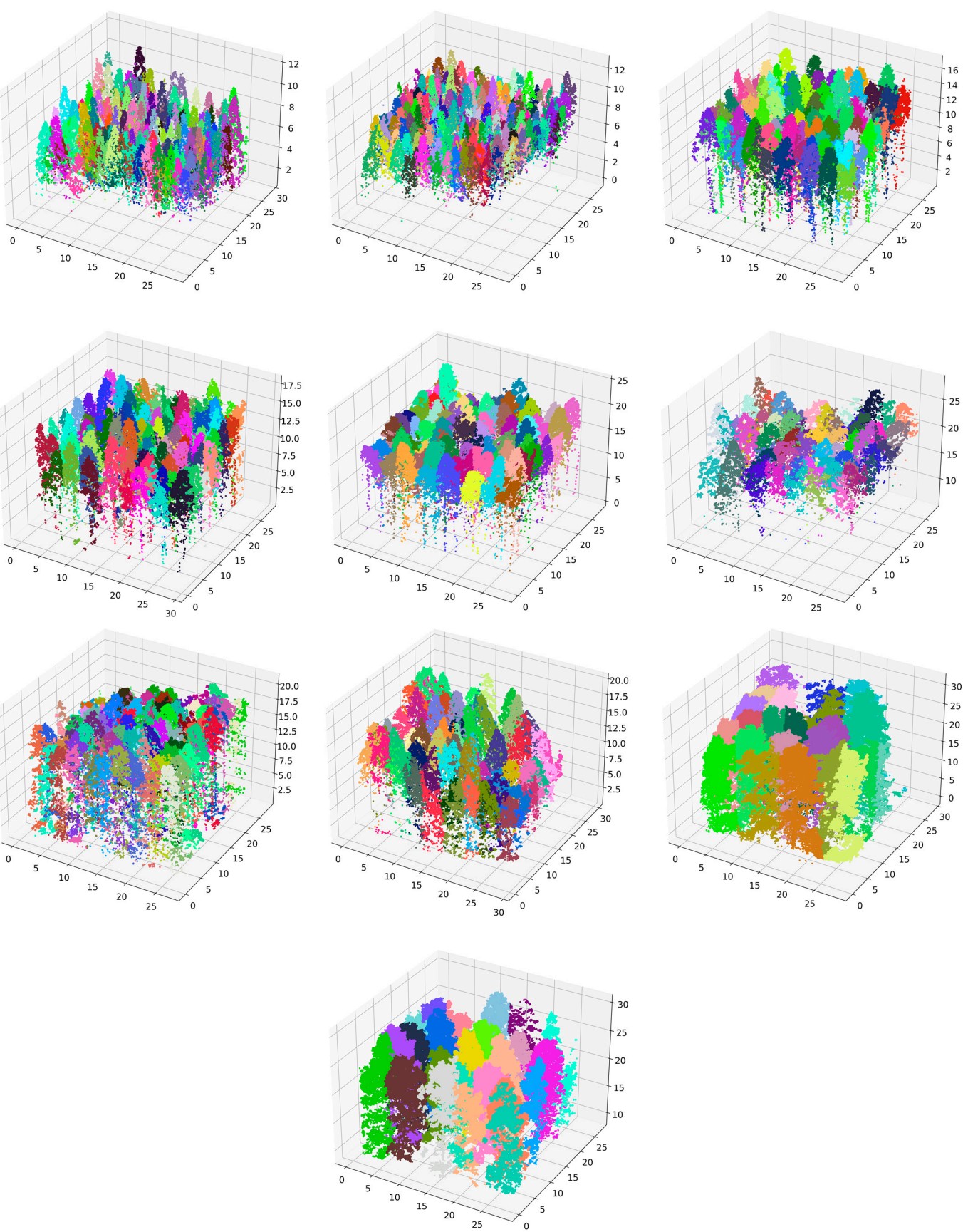

**Figure 6.** Improved K-means segmentation results.

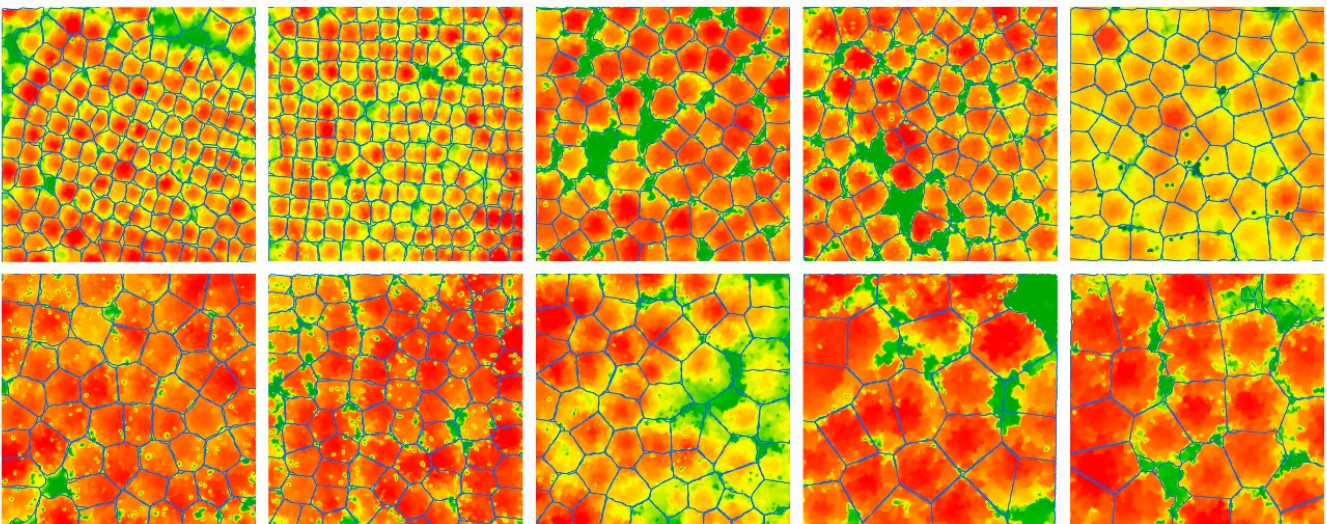

**Figure 7.** The forest boundaries obtained using an alpha-shape algorithm.

**Table 4.** Comparison of single tree segmentation accuracy.

| Segmentation Algorithm | Plot Number | Number of Single Trees with a 1:1 Cor-responding Relationship | Reference Total Number of Single Wood | The Total Number of Single Trees Detected | Producer Accuracy (%) | User Accuracy (%) | Age Group Producer Accuracy (%) | Age Group User Accuracy(%) |
|---|---|---|---|---|---|---|---|---|
| Marked watershed Algorithm | 1 | 132 | 167 | 147 | 79.04 | 89.80 | 74.79 | 85.02 |
| | 2 | 134 | 190 | 167 | 70.53 | 80.24 | | |
| | 3 | 51 | 89 | 80 | 57.3 | 63.75 | 62.54 | 68.19 |
| | 4 | 61 | 90 | 84 | 67.78 | 72.62 | | |
| | 5 | 65 | 78 | 71 | 83.33 | 91.55 | 77.26 | 83.28 |
| | 6 | 42 | 59 | 56 | 71.19 | 75.00 | | |
| | 7 | 35 | 85 | 83 | 41.18 | 42.17 | 46.00 | 47.36 |
| | 8 | 31 | 61 | 59 | 50.82 | 52.54 | | |
| | 9 | 17 | 23 | 23 | 73.91 | 73.91 | 72.37 | 72.37 |
| | 10 | 17 | 24 | 24 | 70.83 | 70.83 | | |
| | Average Value | | | | 66.59 | 71.24 | | |
| Point cloud segmentation Algorithm | 1 | 122 | 167 | 137 | 73.05 | 89.05 | 74.42 | 92.53 |
| | 2 | 144 | 190 | 150 | 75.79 | 96.00 | | |
| | 3 | 64 | 89 | 79 | 71.91 | 81.01 | 77.07 | 86.76 |
| | 4 | 74 | 90 | 80 | 82.22 | 92.50 | | |
| | 5 | 53 | 78 | 70 | 67.95 | 75.71 | 71.27 | 75.15 |
| | 6 | 44 | 59 | 59 | 74.58 | 74.58 | | |
| | 7 | 60 | 85 | 79 | 70.59 | 75.95 | 68.90 | 72.72 |
| | 8 | 41 | 61 | 59 | 67.21 | 69.49 | | |
| | 9 | 20 | 23 | 22 | 86.96 | 90.91 | 85.15 | 90.91 |
| | 10 | 20 | 24 | 22 | 83.33 | 90.91 | | |
| | Average Value | | | | 75.36 | 83.61 | | |
| Traditional K-means Algorithm | 1 | 109 | 167 | 147 | 65.27 | 74.15 | 69.74 | 79.29 |
| | 2 | 141 | 190 | 167 | 74.21 | 84.43 | | |
| | 3 | 49 | 89 | 80 | 55.06 | 61.25 | 65.31 | 71.10 |
| | 4 | 68 | 90 | 84 | 75.56 | 80.95 | | |
| | 5 | 48 | 78 | 71 | 61.54 | 67.61 | 68.91 | 73.99 |
| | 6 | 45 | 59 | 56 | 76.27 | 80.36 | | |
| | 7 | 66 | 85 | 83 | 77.65 | 79.52 | 73.25 | 75.36 |
| | 8 | 42 | 61 | 59 | 68.85 | 71.19 | | |
| | 9 | 13 | 23 | 23 | 56.52 | 56.52 | 61.60 | 61.60 |
| | 10 | 16 | 24 | 24 | 66.67 | 66.67 | | |
| | Average Value | | | | 67.76 | 72.27 | | |
| Improved K-means Algorithm | 1 | 137 | 167 | 147 | 82.04 | 93.20 | 79.44 | 90.32 |
| | 2 | 146 | 190 | 167 | 76.84 | 87.43 | | |
| | 3 | 68 | 89 | 80 | 76.4 | 85.00 | 82.09 | 89.53 |
| | 4 | 79 | 90 | 84 | 87.78 | 94.05 | | |
| | 5 | 57 | 78 | 71 | 73.08 | 80.28 | 77.22 | 83.00 |
| | 6 | 48 | 59 | 56 | 81.36 | 85.71 | | |
| | 7 | 69 | 85 | 83 | 81.18 | 83.13 | 75.02 | 77.16 |
| | 8 | 42 | 61 | 59 | 68.85 | 71.19 | | |
| | 9 | 17 | 23 | 23 | 73.91 | 73.91 | 78.62 | 78.62 |
| | 10 | 20 | 24 | 24 | 83.33 | 83.33 | | |
| | Average Value | | | | 78.48 | 83.72 | | |

The 10 sample plots were divided into 5 age groups: young forest, middle-aged forest, near-mature forest, mature forest, and over-mature forest. The research results showed

that the average producer accuracy of the traditional K-means algorithm improved, and the K-means algorithm, watershed algorithm, and PCS algorithm were 67.76%, 78.48%, 66.59%, and 75.36%, respectively. The average user accuracy of these algorithms were 72.27%, 83.72%, 71.24%, and 83.61%, respectively. The ratio of the number of single trees with a 1:1 corresponding relationship to the reference single tree reflects the completeness of single-tree segmentation, while user accuracy represents the ratio of the number of detected single trees with a 1:1 corresponding relationship to the actual number of single trees, reflecting the correctness of single-tree segmentation. Compared with other single-tree segmentation algorithms, the improved K-means algorithm had the highest accuracy in both producer accuracy and user accuracy. Compared with the traditional K-means algorithm, its producer accuracy increased by an average of 10.72% and its user accuracy increased by an average of 11.45%. This indicates that the improved K-means algorithm was superior to traditional algorithms in both the completeness and accuracy of single-tree segmentation.

By comparing the accuracy of different single-tree segmentation methods in different age groups, it could be seen that when the research sample plot was in the young forest group, the improved K-means algorithm had the highest producer accuracy at 79.44%, and the PCS algorithm had the highest user accuracy at 92.53%. When the research sample plot was in the middle-aged forest group, the improved K-means algorithm had the highest producer accuracy and user accuracy at 82.09% and 89.53%, respectively. When the research sample plot was in the near-mature forest group, the watershed algorithm had the highest producer accuracy and user accuracy, which was slightly higher than the improved K-means method, at 77.26% and 83.28%, respectively. When the research sample plot was in the mature forest group, the improved K-means algorithm had the highest producer accuracy and user accuracy at 75.02% and 77.16%, respectively. When the research sample plot was in the over-mature forest group, the PCS algorithm had the highest producer accuracy and user accuracy at 75.36% and 83.61%, respectively.

To more clearly demonstrate the improvement of the improved K-means algorithm compared to the traditional K-means algorithm, this study used a one-way ANOVA to compare whether there were differences in the segmentation results of different single-tree segmentation algorithms in terms of producer accuracy and user accuracy. In the segmentation results of different single-tree segmentation algorithms, both producer accuracy and user accuracy show significant differences ($p < 0.05$). According to Table 5, the average values of producer accuracy and user accuracy for the improved K-means algorithm were the highest, at 78.47% and 83.72%, respectively. This indicates that the improved K-means algorithm achieves the highest segmentation accuracy compared to the other single-tree segmentation algorithms. In Table 5, the standard deviation and coefficient of variation of producer accuracy and user accuracy for the improved K-means algorithm are the smallest, indicating that the segmentation accuracy of the improved K-means algorithm was more stable under different age conditions compared to the other algorithms. The different lowercase letters in Table 5 indicate significant differences in segmentation accuracy among different single-tree segmentation algorithms ($p < 0.05$), and there were significant differences between the traditional K-means method and the improved K-means method in terms of both producer accuracy and user accuracy. In summary, the improved K-means method showed a significant improvement in single-tree segmentation accuracy compared to the traditional K-means algorithm.

**Table 5.** Descriptive Statistics of Segmentation Accuracy for Different Single-Tree Segmentation Algorithms.

| Segmentation Algorithm | Producer Accuracy (%) | | User Accuracy (%) | |
|---|---|---|---|---|
| | Average ± S.D. | Variation Coefficient | Average ± S.D. | Variation Coefficient |
| Marked watershed Algorithm | 66.59 ± 13.02 [a] | 19.55 | 71.24 ± 15.32 [a] | 21.50 |
| Point cloud segmentation Algorithm | 75.36 ± 6.73 [b] | 8.93 | 83.61 ± 9.29 [b] | 11.12 |
| Traditional K-means Algorithm | 67.76 ± 8.22 [a] | 12.13 | 72.27 ± 9.24 [ab] | 12.79 |
| Improved K-means Algorithm | 78.47 ± 5.66 [a,b] | 7.21 | 83.72 ± 7.31 [b] | 8.74 |

Note: Different lowercase letters indicate significant differences between different single-tree segmentation algorithms (one-way ANOVA, $p < 0.05$).

## 4. Discussion

In this study, we proposed an improved K-means algorithm to address the shortcomings inherent in the traditional K-means algorithm. Combining the results of single-tree segmentation (Table 4) and the results of one-way ANOVA (Table 5), it can be inferred that there was a significant difference ($p < 0.05$) in single-tree segmentation accuracy between the improved K-means algorithm and the traditional K-means algorithm. Compared to the traditional K-means algorithm, the improved K-means algorithm demonstrates better segmentation performance in the single-tree segmentation process of Chinese fir plantations. The main advantages of the method used in this study are as follows: (1) The algorithm used the method of local maximum values to obtain the number and location of trees within the plot range, thereby providing the K value and seed points for the K-means method. The role of the seed points was similar to that of the watershed segmentation algorithm [24] in that it provides an initial position for the algorithm and avoids the impacts of the number and position of initial clustering centers on the segmentation results. (2) Using the height values of the point cloud as weights in the calculation of cluster center positions can promote the movement of cluster centers toward the top of the tree, and in this way avoids the influence of outliers and shifts toward the edges of the tree crown. This contributes to significant improvements in the accuracy and efficiency of single-tree segmentation. However, despite these enhanced features, the improved K-means algorithm also has certain shortcomings. Firstly, in this study, we used the local maximum method to obtain seed points in the digital canopy model, which requires the initial conversion of LiDAR data to depth images. This process is typically cumbersome and can lead to the loss of positional information for some point clouds. Secondly, LiDAR data is classed as a type of big data, and processing dense LiDAR data necessitates considerable time and storage resources, which is not conducive to processing data obtained from large study areas.

According to the results of single-tree segmentation (Table 4), it can be found that the improved K-means method has better performance than other traditional methods in the sample plots of young forest, middle-aged forest, and mature forest. However, in the sample plots of near-mature forest and over-mature forest, its accuracy is inferior to that of the watershed algorithm and PCS algorithm, especially in sample plot 5 of the near-mature forest and sample plot 9 of the over-mature forest. The reason for this may be that sample plot 5 of the near-mature forest had a high degree of canopy closure, and the connection between tree crowns in the sample plot was relatively close. The improved algorithm was difficult to accurately find the edge of the canopy, resulting in a decrease in segmentation accuracy. The watershed algorithm can segment the tree crown according to the gradient change and accurately find the edge of the tree crown, so it has a better segmentation effect. In sample plot 9 of the over-mature forest, there was a large difference in the size of the

single-tree crowns. The improved K-means algorithm will result in over-segmentation when segmenting single trees with larger crown widths, resulting in a decrease in accuracy. The PCS algorithm can perform segmentation based on the distance threshold between tree crowns and was not affected by differences in crown width between trees in the sample plot, so it had a better segmentation effect. According to Table 5, the improved K-means algorithm had the smallest standard deviation and coefficient of variation for both producer accuracy and user accuracy. This indicates that the segmentation accuracy of the improved K-means algorithm was the most stable compared to other segmentation algorithms, and there won't be significant variations in single-tree segmentation accuracy across different age groups. Overall, the improved K-means algorithm can achieve high accuracy in single-tree segmentation results for Chinese fir plantations of different age groups.

In this study, we also found that the insensitivity of the traditional K-means method to point cloud height made it difficult for the algorithm to identify suitable clustering positions, ultimately leading to frequent over-segmentation (Figure 8) [25]. As shown in the figure, the single-tree segmentation result (circled in white) includes part of the crowns of the other trees. By applying the improved K-means algorithm, we were able to significantly improve the accuracy of single-tree segmentation. We suspect that this effect may be due to the local maximum method, which allows the improved K-means algorithm to identify appropriate initial clustering centers. In addition, height weighting sensitivity helps to maintain clustering near the tree top, reducing the occurrence of over-segmentation. This phenomenon is similar to the watershed method, which often suffers from over-segmentation, resulting in the same target being divided into multiple objects and losing the original features of the segmentation target. The marker watershed method adds marker points to avoid over-segmentation of a target [26]. The initial clustering points and height weighting added in this study have a similar effect to marker points, avoiding the occurrence of over-segmentation in classification results. In addition, this phenomenon often occurs in the PCS algorithm, which performs clustering segmentation on all point clouds below the tree top by setting a distance threshold without setting fixed clustering centers and point cloud weights, relying only on distance threshold judgment, resulting in frequent under-segmentation or over-segmentation. Adding marker points to the PCS algorithm may reduce its over-segmentation or under-segmentation. Figure 9 shows a comparison of the segmentation results of the traditional and improved K-means algorithms. From the position circled in white, it can be seen that using the improved K-means algorithm can correctly segment single trees in this area, while using the traditional K-means algorithm cannot. Comparing the performance of the two algorithms shows that although the optimized K-means algorithm is significantly better in segmentation accuracy for young and middle-aged forest groups, its performance improvement is not significant when applied to mature forest groups, where we did not see an improvement in single-tree segmentation accuracy. This may be because in mature forests, trees are densely arranged, and differences in tree height are usually relatively small. In summary, the improved K-means method can effectively improve the accuracy of single-tree segmentation.

On the basis of our study of cloth filtering, we found that under the same flight parameters, canopy density has a considerable influence on the efficacy of LiDAR penetration. Under conditions of high canopy density, LiDAR penetration is poor, thereby resulting in a reduction of the number of points radiating to the ground and, consequently, in the displacement of tree top positions and errors in tree height determinations. This phenomenon has similarly been observed by other researchers, such as Liu [27]. Accordingly, in future studies, UAV-LiDAR and TLS (Terrestrial Laser Scanning) point cloud data could be combined to extract tree top points. Given that UAV-LiDAR provides a complete canopy structure, whereas TLS provides complete ground height information, a combination of these two approaches could be applied to achieve higher-precision normalized point cloud data. This would thereby further improve the accuracy of tree top position and tree height determinations, and thus enhance the overall accuracy of single-tree segmentation.

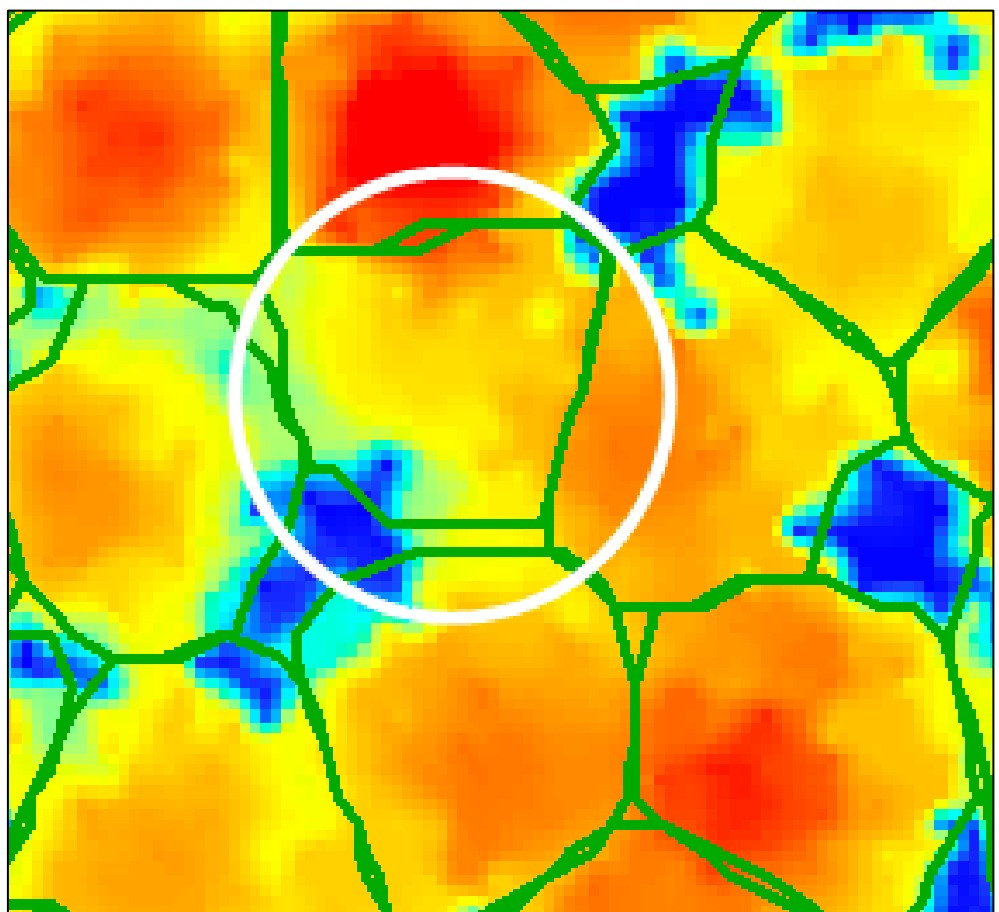

**Figure 8.** The traditional K-means algorithm exhibits the phenomenon of over-segmentation. Note: The white circles in the figure represent false segmentation.

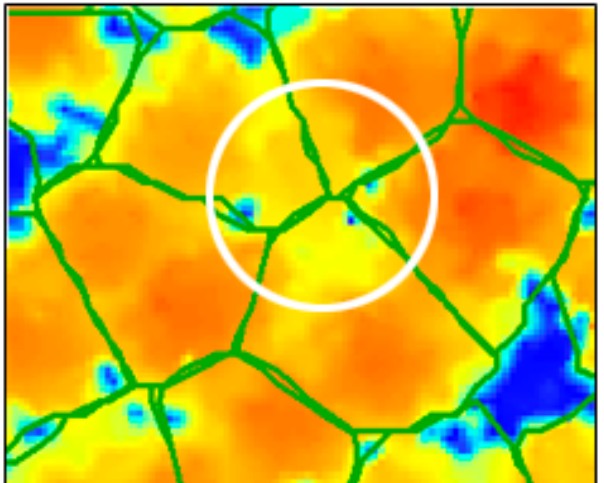 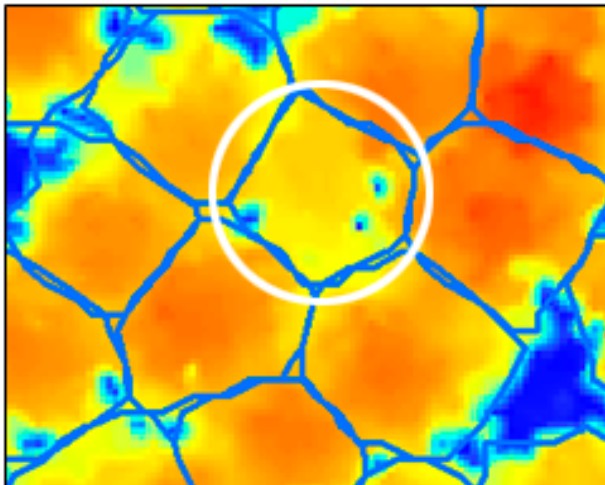

**Figure 9.** Comparison of single tree segmentation results between the traditional K-means algorithm (**left**) and the improved K-means algorithm (**right**).

This study algorithm mainly focused on Chinese fir plantations and did not attempt to segment other types of tree species. In the future, the segmentation accuracy of this algorithm can be compared when applied to different tree crown shapes. In addition, this algorithm currently only focuses on pure forests and has not considered the use of mixed forests, which also places higher demands and challenges on the algorithm. In

future research, more consideration can be given to these complex situations to increase the algorithm's general applicability.

## 5. Conclusions

This study proposed an improved K-means algorithm for single-tree segmentation based on high-density LiDAR point-cloud data obtained by unmanned aerial vehicles. Compared with the traditional K-means algorithm, PCS algorithm, and marked watershed algorithm, the segmentation accuracy of single tree in different age groups of Chinese fir plantations significantly improved. This research showed that the improvement effect of the improved K-means algorithm compared to the traditional K-means algorithm varied among different age groups. The greatest improvement effect was observed in young and middle-aged forests, followed by over-mature forests, whereas the improvement effect in near-mature and mature forests was less pronounced. The segmentation accuracy of the improved K-means algorithm was more stable compared to other segmentation algorithms across different age groups and could achieve high accuracy in single-tree segmentation results for Chinese fir plantations of different age groups.

This research showed that the improved K-means algorithm needed to use LiDAR point cloud data to construct a CHM image and then use the local maximum method to obtain the local maximum of the CHM to determine the position of the initial clustering center and the clustering K value for the K-means algorithm. Although this step can accurately obtain the parameters required for the improved K-means algorithm, the process is cumbersome and is likely to cause data loss during the conversion of LiDAR data into 2D data. This can affect the recognition accuracy of the maximum value point. Therefore, it is worth exploring whether it is possible to directly extract the initial clustering center and K value from height-normalized point-cloud data. Given that the traditional K-means algorithm and its derivative algorithms need to iterate over the entire sample dataset several times until the clustering result converges or reaches the maximum iteration set, such methods require a substantial amount of computational resources and storage space when dealing with large-sample data. LiDAR point-cloud data collected by laser radar fall within the category of big data. Therefore, using the K-means algorithm to process point-cloud data can be time consuming. Therefore, future research can reduce the amount of computation by diluting and voxelizing point-cloud data and improving the efficiency of point-cloud segmentation.

The results of the cloth filtering algorithm have shown that under the same flight parameters, forest canopy density has a significant impact on the penetration effect of LiDAR. The penetration ability of LiDAR is poor in high forest canopy density conditions. Simultaneously, in the simulated terrain flight mode, the energy consumption of small, unmanned aerial vehicles is relatively fast, and the research area that can be covered in a single flight mission is limited. Compared with traditional aircraft, their operational efficiency is relatively low. Its performance and industry suitability need to be further improved when applied to southern hilly areas with large terrain changes.

In summary, the emergence of UAV LiDAR point-cloud data has provided a new direction for forestry remote-sensing research. Based on LiDAR point-cloud data, the single-tree segmentation algorithm proposed in this study overcomes the shortcomings of the traditional K-means algorithm and can obtain high-precision single-tree segmentation results. This can provide rich spatial structure information at the single-tree scale and improve the efficiency of forest structure parameter estimation.

**Author Contributions:** Conceptualization, X.C. and K.Y.; Methodology, X.C.; Software, X.C., S.Y. and Z.H.; Validation, X.C.; Formal analysis, X.C., S.Y., Z.H. and H.T.; Investigation, S.Y., H.T., Y.C. and X.H.; Data curation, S.Y., Z.H., H.T. and Y.C.; Writing—original draft, X.C.; Writing—review & editing, X.C.; Supervision, K.Y. and J.L.; Project administration, K.Y. and J.L.; Funding acquisition, K.Y. All authors have read and agreed to the published version of the manuscript.

**Funding:** This research was funded by the National Natural Science Foundation Project (grant number 32271876) and Research on Key technologies of intelligent monitoring and carbon sink metering of forest resources in Fujian Province (grant number 2022FKJ03).

**Data Availability Statement:** The data that support the findings of this study are available from the author upon reasonable request.

**Conflicts of Interest:** The authors declare no conflict of interest.

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
