# Peer review of "Study on Single-Tree Segmentation of Chinese Fir Plantations Using Coupled Local Maximum and Height-Weighted Improved K-Means Algorithm"

_forests, doi:10.3390/f14112130_

Round 1
Reviewer 1 Report
Comments and Suggestions for Authors
I have now completed the review of the study entitled “Study on Single-tree Segmentation of Chinese Fir Plantations using Coupled Local Maximum and Height-weighted Improved K-means Algorithm”. The study is well-written, well-structured and well-communicated. These types of studies are very important in the frame of forest management. However, I believe that the results do not fully support the findings of the study. My comments are provided in the following lines.
General comments
According to the authors “Compared with the traditional K-means algorithm, the algorithm proposed in this study improved the producer accuracy and user accuracy by 10.72% and 11.46%, respectively; The segmentation accuracy of this study's algorithm was significantly improved in the two age groups of young and middle-aged forests, but its performance was slightly inferior in mature forests”. The outcomes are solely based on descriptive statistics comparison of the tested samples. However, the outcomes must be referred at the population level. Therefore, I strongly recommend using the appropriate statistical tests in order to detect statistically significant differences between the two methods so as to increase the scientific merit of the ms.
Specific comments
- The abstract must be shortened to 200 words maximum, according to the journal’s recommendations.
- I suggest increasing the figure’s resolution throughout the ms.
- L102 How the canopy cover was estimated?
- L136 I suggest removing the “however” word.
- L141 Please, remove the parenthesis.
- L196 Please, improve the expression.
Author Response
Responds to the reviewer’s comments: Reviewer 1:
- The abstract must be shortened to 200 words maximum, according to the journal’s recommendations.
Response: We appreciate the valuable suggestion. As you suggested, we have reduced the length of the abstract to less than 200 words.
- I suggest increasing the figure’s resolution throughout the ms.
Response: We apologize for the low resolution of the figure, now we have improved the resolution of the image.
- L102 How the canopy cover was estimated?
Response: We used the following method to estimate the canopy density. Two linear transects were set along the diagonal of the plot, with a total of 100 sampling points. Each sampling point was observed upward. If it was observed to be covered by vegetation, it was recorded as canopy covered point, otherwise, it was recorded as blank point. Finally, the canopy cover of the plot was obtained by dividing the total number of canopy covered points by the number of all points.[ Meng, X.-Y. Forest Measuration; Chinese Forestry Press: Beijing, China, 2007; pp. 10–11.]
- L136 I suggest removing the “however” word.
Response: Thanks for your suggestion, we have removed the word " however ".
- L141 Please, remove the parenthesis.
Response: Based on your suggestion, we have removed the parenthesis and rewritten the sentence.
- L196 Please, improve the expression.
Response: Thanks for your valuable suggestions, we have rewritten the paragraph to enhance its expression and readability.
Reviewer 2 Report
Comments and Suggestions for Authors
The authors of the manuscript entitled “Study on Single-tree Segmentation of Chinese Fir Plantations using Coupled Local Maximum and Height-weighted Improved K-means Algorithm” present an interesting approach for tree segmentation in a Chinese fir plantation using the K-means algorithm. Overall, the manuscript is well structured and it is apparent that a lot of effort has been put into this research. Nevertheless, the text is not well-written and often hard to follow due to the poor use of the English language in most parts of the manuscript and the ambiguity of the overall workflow. More specifically:
Many references are missing, especially in the introduction (l. 52,61,71)
Why did the authors choose to compare the specific method with kmeans and not watershed or other well-established algorithms?
Fir plantations are a rather easy task for segmentation, even for the simplest chm-based methods. In that regard, the authors should highlight the novelty.
L.140-141 The line about the CSF algorithm should be rewritten.
L. 149 What is the “laser radar rasterisation method”? According to the description, I could guess that Kriging is the proposed algorithm.
Overall, the methodology section is really difficult to follow. The authors constructed the DSM and DEM in order to create the CHM, which could be easily constructed from the point cloud directly. It is really difficult to find any reasonable explanation behind this decision.
L. 168 The tree tops are detected in the CHM, which is normalized. In steep slopes, like your study area, the tree top can be shifted due to the ground effect. Did you make any process to remove the slope effect?
Your approach is fully based on the tree tops, which are detected by the LMF. However, this filter has several flaws, especially in multilayered forests. How did the authors address this issue?
The k-means clustering process is unclear and hard to follow. The authors should completely revise the specific section.
Sections 3.1 and 3.2 should be removed. Both sections present a standard procedure with no experimental results.
Last, Discussion is hard to follow, especially between l.371 and 392. The authors should re-write this section, providing a comparison between previous/similar studies and more insights about their findings.
Comments on the Quality of English Language
Moderate editing of English language required
Author Response
1.Many references are missing, especially in the introduction (l. 52,61,71)
Response: We appreciate the valuable suggestion. As you suggested, we have added references to these sections.
- Why did the authors choose to compare the specific method with kmeans and not watershed or other well-established algorithms?
Response: Appreciate your feedback. On the basis of the original experiment, we add two well-established segmentation algorithms, watershed and PCS, to compare the segmentation accuracy.
- Fir plantations are a rather easy task for segmentation, even for the simplest chm-based methods. In that regard, the authors should highlight the novelty.
Response: Thanks for the suggestion. The young Chinese fir forest is arranged orderly in the artificial forest environment, and the crown shape is uniform. However, with the gradual increase of age, the Chinese fir plantation is affected by selective felling, sanitation felling and other forest management measures, so the arrangement of trees is no longer orderly, and the shape of Chinese fir crown tends to be irregular, which brings certain challenges to the accurate segmentation of the tree crown. At the same time, mature forest of Chinese fir is an important wood reserve resource and the core of forest carbon pool, so it is of great value to accurately monitor mature forest and over-mature forest.
- L.140-141 The line about the CSF algorithm should be rewritten.
Response: Based on your suggestion, we have removed the parenthesis and rewritten the sentence.
- L. 149 What is the “laser radar rasterisation method”? According to the description, I could guess that Kriging is the proposed algorithm.
Response: We appreciate the valuable suggestion. “Laser radar rasterisation method” is a tool in ArcGIS Pro 3.1, its essence is indeed an interpolation method, and Kriging method is also one of the commonly used interpolation methods for constructing DEM, but different from Kriging method, our study uses the triangulation interpolation methods to construct DEM, the method constructs triangulation irregular network to obtain the basic pixel value, and then uses Natural Neighbors method to interpolate the blank pixel value.
- Overall, the methodology section is really difficult to follow. The authors constructed the DSM and DEM in order to create the CHM, which could be easily constructed from the point cloud directly. It is really difficult to find any reasonable explanation behind this decision.
Response: Thanks for your valuable suggestions. Canopy height model is a model that represents the height of tree canopy surface, the height model built directly with the point cloud is actually a digital surface model, the pixel value represents the sum of terrain height and tree canopy height. In order to make the raster data set represent only the canopy height value, it is necessary to construct the digital elevation model and the digital surface model representing the ground height respectively, and subtract the pixel value of the DSM from the pixel value of the DEM to obtain the canopy height model representing only the canopy height information.
- L. 168 The tree tops are detected in the CHM, which is normalized. In steep slopes, like your study area, the tree top can be shifted due to the ground effect. Did you make any process to remove the slope effect?
Response: Thanks for the suggestion, some scholars on this issue pointed out that after normalization treatment, the tree apex will indeed be offset, but depending on the crown shape, the slope of the fir crown is large, the treetop area is more concentrated, and the offset will become small. In addition, K-means as a clustering algorithm, as long as the position of the tree vertex is near the top of the tree, the algorithm will react to it and segment the point cloud. Considering the actual situation of the sample plot in this study, the deviation of the tree vertex has a small impact on the experimental results of this study and can be ignored.
- Your approach is fully based on the tree tops, which are detected by the LMF. However, this filter has several flaws, especially in multilayered forests. How did the authors address this issue?
Response: Thanks for the suggestion. The sample land used in this study belongs to Chinese fir plantation. The trees in the same plot have similar height, and the understory is relatively flat and clean, so there is no multilayered forest.
- The k-means clustering process is unclear and hard to follow. The authors should completely revise the specific section.
Response: Thanks for your valuable suggestions, we have rewritten the paragraph to enhance its expression and readability.
- Sections 3.1 and 3.2 should be removed. Both sections present a standard procedure with no experimental results.
Response: Based on your suggestion, we removed sections 3.1 and 3.2.
- Last, Discussion is hard to follow, especially between l.371 and 392. The authors should re-write this section, providing a comparison between previous/similar studies and more insights about their findings.
Response: Thanks for your valuable suggestions, we rewrote our discussion section, adding other similar studies to the discussion section, and analyzed their findings in conjunction with our research content.
Round 2
Reviewer 1 Report
Comments and Suggestions for Authors
The authors have improved the quality of the ms, but the main problem still remains. I strongly recommend implementing a simple statistical method (e.g. one-way anova) to export results at the population level using p-values. Again, I believe a simple means comparison is insufficient for conclusions.
Author Response
- The authors have improved the quality of the ms, but the main problem still remains. I strongly recommend implementing a simple statistical method (e.g. one-way ANOVA) to export results at the population level using p-values. Again, I believe a simple means comparison is insufficient for conclusions.
Response: Thank you for your valuable suggestions. We have made further improvements to the image quality and have uploaded all the original images mentioned in the paper as attachments in a folder. Additionally, we have followed your suggestion and used one-way ANOVA to determine the statistical significance differences among multiple segmentation methods, thereby enhancing the scientific value of our conclusions.
Reviewer 2 Report
Comments and Suggestions for Authors
The authors answered the majority issues and provided an improved manuscript. I have only two comments:
1)The laser radar rasterization process is not a method, but a tool in ArcPRO. So you have to describe all the parameters used and try to avoid using "method", it is a tool.
2)Regarding Comment 6 from the previous round, CHM can be easily extracted directly from the point cloud. The rasterization of the normalized point cloud (using pit free or other similar algorithms) will provide a CHM, so the whole process described in Section 2.3.3 is irrelevant. If the ground points are labeled as ground, you can directly rasterize the terrain and normalize your point cloud. In my opinion, you should remove this Section and add the CHM construction after the point cloud normalization. It is considered as a standard LiDAR processing routine and may confuse the readers.
Comments on the Quality of English Language
Minor editing of English required
- The laser radar rasterization process is not a method, but a tool in ArcPRO. So you have to describe all the parameters used and try to avoid using "method", it is a tool.
Response: Thank you for your valuable suggestion. We have made modifications to the textual expression of the tool in the paper and have described all the parameters used.
- Regarding Comment 6 from the previous round, CHM can be easily extracted directly from the point cloud. The rasterization of the normalized point cloud (using pit free or other similar algorithms) will provide a CHM, so the whole process described in Section 2.3.3 is irrelevant. If the ground points are labeled as ground, you can directly rasterize the terrain and normalize your point cloud. In my opinion, you should remove this Section and add the CHM construction after the point cloud normalization. It is considered as a standard LiDAR processing routine and may confuse the readers.
Response: Appreciate your feedback. We have removed the original expression in section 2.3.3 and modified it to describe the construction of CHM using normalized point cloud.
Round 3
Reviewer 2 Report
Comments and Suggestions for Authors
The authors successfully replied to all of my points, so I have no further comments.